# Visual experience has opposing influences on the quality of stimulus representation in adult primary visual cortex

Brian B Jeon[1,2,3], Thomas Fuchs[2,3,4], Steven M Chase[1,2,3], Sandra J Kuhlman[1,2,3,4]*

[1]Department of Biomedical Engineering, Carnegie Mellon University, Pittsburgh, United States; [2]Center for the Neural Basis of Cognition, Carnegie Mellon University, Pittsburgh, United States; [3]Neuroscience Institute, Carnegie Mellon University, Pittsburgh, United States; [4]Department of Biological Sciences, Carnegie Mellon University, Pittsburgh, United States

**Abstract** Transient dark exposure, typically 7–10 days in duration, followed by light reintroduction is an emerging treatment for improving the restoration of vision in amblyopic subjects whose occlusion is removed in adulthood. Dark exposure initiates homeostatic mechanisms that together with light-induced changes in cellular signaling pathways result in the re-engagement of juvenile-like plasticity in the adult such that previously deprived inputs can gain cortical territory. It is possible that dark exposure itself degrades visual responses, and this could place constraints on the optimal duration of dark exposure treatment. To determine whether eight days of dark exposure has a lasting negative impact on responses to classic grating stimuli, neural activity was recorded before and after dark exposure in awake head-fixed mice using two-photon calcium imaging. Neural discriminability, assessed using classifiers, was transiently reduced following dark exposure; a decrease in response reliability across a broad range of spatial frequencies likely contributed to the disruption. Both discriminability and reliability recovered. Fixed classifiers were used to demonstrate that stimulus representation rebounded to the original, pre-deprivation state, thus dark exposure did not appear to have a lasting negative impact on visual processing. Unexpectedly, we found that dark exposure significantly stabilized orientation preference and signal correlation. Our results reveal that natural vision exerts a disrupting influence on the stability of stimulus preference for classic grating stimuli and, at the same time, improves neural discriminability for both low and high-spatial frequency stimuli.

*For correspondence: skuhlman@cmu.edu

Competing interest: The authors declare that no competing interests exist.

## Editor's evaluation

The present manuscript examines cortical representations of basic visual attributes following a manipulation shown to enhance plasticity in the adult brain: binocular dark exposure for 8 days, followed by light re-introduction. Prior work did not rule out the possibility that prolonged dark exposure could negatively impact visual representations in V1. Using 2P calcium imaging in awake adult mice to quantify changes in stimulus selectivity, discriminability, and reliability of V1 neurons, Jeon and colleagues provide compelling evidence that dark exposure has opposing but transient effects at the single neuron versus population level, thus failing to permanently disrupt visual representations in V1.

## Introduction

Sensory cortex is highly malleable early in life. During postnatal development, cortical territory rapidly expands and contracts to represent active and inactive inputs, respectively. These large-scale changes are mediated by age-restricted experience-dependent synaptic refinement at the level of individual postsynaptic neurons (*Buonomano and Merzenich, 1998*). Primary visual cortex (V1) is particularly sensitive to visual deprivation during this period. Classic experiments demonstrate that in order for sensory pathways to drive cortical responses in the adult, continuous binocular input is required during postnatal development (*Reh et al., 2020*; *Hensch and Quinlan, 2018*). Experimentally induced monocular deprivation in young animals results in a rapid decrease in the strength of deprived inputs, and with a delay, the response of intact inputs is potentiated. Depression of deprived inputs occurs at a synaptic level and outlasts the perturbation. On the other hand, the same peripheral perturbation in adults does not induce a rapid decrease in response strength, although potentiation of the intact input does proceed (*Fong et al., 2021*; *Jenks et al., 2017*). Furthermore, in the young, the impact of deprivation on excitatory neuron activity levels is compensated for by a decrease in evoked inhibitory neuron activity. This form of compensation is a direct consequence of deprivation and does not occur in adults (*Feese et al., 2018*). Delineating the molecular basis and cellular signaling pathways responsible for restricting plasticity in the adult is an active area of investigation. For example, consistent with the above observations, extracellular matrix perineuronal nets (PNNs), which preferentially surround parvalbumin inhibitory neurons and create a barrier for synaptic remodeling, are resistant to degradation in the adult. Interventions that target the break-down of PNNs in adult V1 are effective in restoring juvenile-like plasticity and allow previously unused inputs to regain the ability to drive cortical neurons (*Reh et al., 2020*; *Murase et al., 2019*; *Faini et al., 2018*; *Jenks et al., 2021*; *Pizzorusso et al., 2006*; *Pizzorusso et al., 2002*). Taken together, there is a general consensus that in the adult, the inputs that are established during postnatal development retain a limited amount of plasticity throughout life. However, in contrast to the young, established inputs are not lost following disuse, and in the absence of additional training or treatment, new input patterns from previously occluded sources are not readily integrated into the existing networks of adults (*Hensch and Quinlan, 2018*). As such, restoring vision to subjects that have matured without binocular input during early postnatal development is a recognized challenge (*Hensch and Quinlan, 2018*; *Falcone et al., 2021*; *Rodríguez et al., 2018*).

To successfully restore vision, the newly opened inputs must compete with non-deprived eye inputs to drive cortical responses. An emerging treatment for animals with experimentally induced amblyopia, across a diverse range of species, is to transiently expose subjects to darkness, followed by light reintroduction (LRx; *Eaton et al., 2016*; *He et al., 2006*; *Montey and Quinlan, 2011*; *Murase et al., 2017*; *He et al., 2007*; *Erchova et al., 2017*; *Stodieck et al., 2014*; *Mitchell et al., 2019*). Accumulating evidence indicates that dark exposure (DE) followed by LRx re-activates juvenile plasticity and is sufficient to restore deprived-eye input responses to grating stimuli. Notably, LRx effectively degrades PNNs (*Murase et al., 2017*). Thus, this critical form of juvenile plasticity is reactivated and likely contributes to the effectiveness of DE in restoring responsiveness of the deprived pathway (*Murase et al., 2019*).

Ideally, treatments such as DE followed by LRx would not have a negative impact on on-going visual processing carried out by the intact pathway. In other words, effective treatments would facilitate the integration of new information without perturbing existing functionality. Based on the studies cited above, it is not expected that in the adult basic responsiveness to visual stimuli following transient DE would be lost. However, previous work demonstrated that closing one eye is sufficient to transiently disrupt stimulus representation in the adult. Although individual neurons remain responsive, the pattern of activity evoked in V1 is disturbed, including orientation preference and pairwise signal correlation among simultaneously recorded neurons (*Rose et al., 2016*). Furthermore, although mean firing rates are largely similar across daily light-dark transitions, pairwise correlations are significantly stronger during vision (*Torrado Pacheco et al., 2019*). This raises the possibility that deprivation such as DE could increase the rate of representational drift (*Deitch et al., 2021*), placing a burden on downstream areas to update the manner in which information is readout. Indeed, in the olfactory cortex, continuous experience is required for drift to remain low (*Schoonover et al., 2021*). Furthermore, higher-order relationships among neurons, beyond the basic responsiveness of individual neurons (*Rupasinghe et al., 2021*), could be altered by the interruption of continuous visual input.

To determine whether transient DE disrupts or otherwise influences stimulus representation beyond basic responsiveness, neural responses to grating stimuli that covered a broad range of spatial frequencies were recorded using two-photon calcium imaging in awake head-fixed mice before and after transient DE. Calcium imaging has the advantage that individual neurons can be readily longitudinally tracked (*Margolis et al., 2012*). Tuning stability and response reliability were assessed before and after DE. In addition, neural discriminability and the rate of representational drift were quantified using classifiers. We found that in contrast to monocular deprivation (*Rose et al., 2016*), DE did not degrade pairwise signal correlation when immediately assessed after DE and in fact stabilized orientation preference in neurons that remained tuned. Similarly, LRx did not degrade pairwise signal correlation of tuned neurons. These data indicate that changes in signal correlation induced by monocular deprivation in adulthood are likely a result of imbalanced input rather than reduced drive from the periphery.

However, similar to olfactory cortex, an increase in the rate of representation drift was detected when visual input was interrupted by DE. A decrease in the trial-to-trial reliability of stimulus responsiveness accounted for the transient change. Reliability was restored within 8 days of LRx, and the representation rebounded to its original form. Thus when used for the treatment of amblyopia, neither DE nor LRx is expected to have a persistent negative impact on existing visual processing, although the effectiveness of perceptual training (*McGuire et al., 2022*) immediately following DE may be influenced by a transient increase in representational drift. Furthermore, our results establish that although natural vision, which includes complex scene statistics, has a disrupting influence on tuning stability to simple grating stimuli, natural vision as experienced in the home-cage environment improves neural discriminability in the adult.

## Results and discussion

To assess the impact of 8 days of DE on tuning stability and neural discriminability, the activity of individual layer 2/3 excitatory V1 neurons in the binocular zone was imaged in response to randomized presentations of static grating stimuli using two-photon microscopy in head-fixed transgenic mice positioned atop a floating spherical treadmill, expressing the calcium indicator GCaMP6f driven by the EMX1 promoter. Our goal was to assess the stability of stimulus representation before and after DE. Therefore prior to DE, two baseline imaging sessions acquired 8±1 days apart, referred to here as Baseline 1 (B1) and Baseline 2 (B2) were recorded. The acquisition of two baseline sessions allowed the stability of tuning to grating stimuli of varying orientation and spatial frequency (s.f.) to be assessed before DE was initiated. A third imaging session was recorded immediately after DE and is referred to as the post-DE (pDE) session. A final forth imaging session was acquired after 8±1 days of LRx, referred to as the recovery (Rec) session. These four sessions were used to define three experimental conditions: control, DE, and LRx (*Figure 1A and B*). To facilitate interpretation of stability, the elapsed time in-between imaging sessions was held constant. A total of six mice was included in the study (see *Table 1* for sex and age information). The fraction of visually responsive neurons, defined as those neurons whose activity was significantly modulated by stimulus feature (ANOVA, $\alpha=0.05$; see Methods), is reported in *Supplementary file 1A* for the six mice in each of the imaging sessions. As expected, 30–50% of the segmented neurons were responsive to full-field static gratings (*Ohki et al., 2005*; *Montijn et al., 2016*; *Ko et al., 2014*; *Jeon et al., 2018*; *de Vries et al., 2020*). Locomotion and pupil diameter were monitored; trials in which locomotion or eye blinking was detected were removed from analysis.

### Orientation preference of well-tuned neurons is stabilized by DE

Tuning to visual stimuli was assessed by fitting deconvolved responses of visually responsive neurons to a two-dimensional Gaussian function, on a trial-by-trial basis (*Jeon et al., 2018*). This method takes into account trial-to-trial variability and ensures that the stimulus identities that elicit activity are more consistent across trials than expected by chance. The stimulus set consisted of 12 orientations and 15 spatial frequencies spanning a range of 0.02–0.30 cycles/°, resulting in a total of 180 stimuli (*Figure 1C*). Four parameters were computed from the two-dimensional fits: orientation preference and bandwidth, as well as s.f. preference and bandwidth. Stability of each of the four parameters was calculated by comparing the absolute change (Δ) during the control and DE experimental conditions.

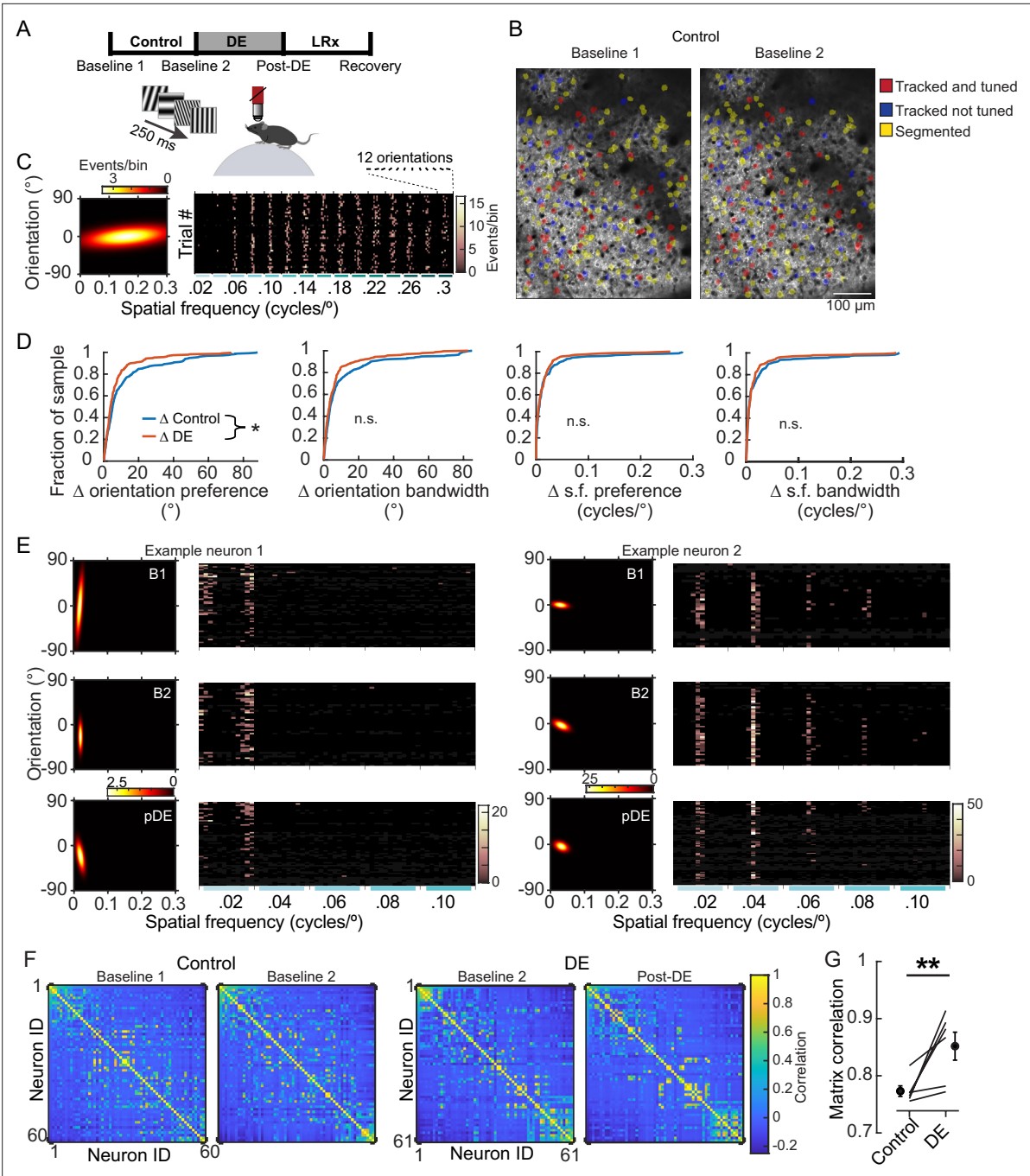

**Figure 1.** 8 days of dark exposure (DE) stabilized orientation preference in V1 neurons. (**A**) Experimental design; three conditions were studied: control, DE, and light reintroduction (LRx), bounded by four imaging sessions, as indicated. Imaging sessions were acquired, while static gratings were presented to awake, head-fixed mice, 8±1 days apart. 12 orientations and 15 spatial frequencies, ranging from 0.02 to 0.3 cycles/°, were presented. (**B**) Example of longitudinal imaging across two sessions. (**C**) Example neuron tuning curve (left) and trial responses to 180 stimuli (right). Stimulus presentations were sorted post-hoc. (**D**) The change in four features, as indicated, is plotted for individual neurons and pooled across six animals. Orientation preference was significantly less stable in the control condition (Wilcoxon rank-sum test corrected for four multiple comparisons [z: 2.467], p=0.032; approximately 30% of the population is shifted leftward in the DE condition) in the control condition (n=249 neurons) compared to the DE condition (n=230 neurons). All neurons that were tracked and tuned to grating stimuli on both sessions were included in the analysis. The same set of neurons was used for a given condition but could differ across conditions to maximize the number of neurons tracked. Statistics for the other three parameters, left to right: Wilcoxon rank-sum test corrected for four multiple comparisons (z: 2.213, 1.478, and 0.5596, power$_e$: 0.711, 0.317, and 0.255), p=0.0538, 0.186, and 0.576. (**E**) Response profiles of two example neurons (tuning curves, left, and trial responses cropped to 0.1 cycles/°, right) are shown for the Baseline 1 (**B1**), Baseline 2 (**B2**), and post-DE (**pDE**) imaging sessions. (**F**) Example of signal correlation matrices from one animal

*Figure 1 continued on next page*

*Figure 1 continued*

comparing the B1 and B2 imaging sessions, and the B2 and Post-DE imaging sessions. For visualization of correlation structure, neuron #1 was randomly selected, and the remaining neurons were sorted in descending order. The same sort-matrix was applied to both sessions. (**G**) Similarity of signal correlation matrices for a given animal (computed as the Pearson's correlation coefficient between the two signal correlation matrices) was significantly higher in the DE condition compared to the control condition (Wilcoxon rank-sum test [*ranksum: 23*], p=0.009, n=6 animals). All tracked and tuned neurons were included, as in 'D'. * p<0.05 and **p<0.01.

The online version of this article includes the following figure supplement(s) for figure 1:

**Figure supplement 1.** Assessment of the effect of dark exposure (DE) on response stability was not influenced by how well individual neurons were fit by the two-dimensional Gaussian function.

**Figure supplement 2.** Strategy to target the imaging field of view to binocular V1.

We took a conservative approach and only considered neurons that were tracked and tuned (i.e. well-fit by the two-dimensional Gaussian function, see Methods for details) on both of the imaging sessions used to calculate the change. DE induced a significant shift in the stability of orientation preference (Wilcoxon rank-sum p=0.032, *Figure 1D and E*). Analysis at the level of individual animals revealed that five out of the six mice reflected the pooled population of neurons (*Figure 1—figure supplement 1A*). The age at the time of the first imaging session ranged from p45 to p86. Notably, the atypical animal was the youngest, p45. Prior work establishes that some rejuvenating influences of DE do not occur until on or after p55 (*Huang et al., 2010*), therefore age might be a contributing factor as to why this animal was distinct.

The detected difference in orientation stability did not appear to be a result of a systematic difference in the quality of the two-dimensional Gaussian fits across the conditions, for the following reasons. First, goodness of fit values, calculated on a trial-by-trial basis for each neuron included in the analysis (see Methods), was not different within or across the two conditions (*Figure 1—figure supplement 1B*; see also the associated file of all tuning curves for neurons scored as being significantly tuned). Second, further analysis in which the top 10% of the worst-fit neurons was removed also revealed an increase in orientation tuning stability in the DE condition (*Figure 1—figure supplement 1C*). We did however note that there was an association between orientation stability and the goodness of fit values. This relationship could have a biological basis. For example, higher trial-to-trial variability could be predictive of a neuron being more likely to shift its orientation preference (*Figure 1—figure supplement 1D*).

The other three parameters trended in the same direction as orientation preference, but the difference between conditions did not reach statistical significance. Given the trend across the four parameters, it is possible that if all four parameters were considered simultaneously, a clear difference would emerge. To address this possibility, we examined pairwise signal correlation among the neurons that were responsive and tuned (*Figure 1F*). The similarity of signal correlation in the control condition and in the DE condition was computed for each of the six animals. We found that the similarity was significantly higher in the DE condition (Wilcoxon rank-sum test, p=0.009; *Figure 1G*). As above, additional analysis was performed to assess whether the results were robust to issues related to the quality of the tuning fits. Signal correlation and the similarity of signal correlation matrices were computed using all responsive neurons, regardless of whether tuned or untuned. Similarity was also significantly higher in

**Table 1.** Animal information including age at each imaging session.

| Animal ID | Animal label | Sex | Age at time of imaging, days old (days in between sessions) | | | |
|---|---|---|---|---|---|---|
| | | | Baseline 1 | Baseline 2 | Post-DE | Recovery |
| 2452_1R | Mouse 1 | M | 83 | 90 (7) | 98 (8) | 106 (8) |
| 2452_1R1L | Mouse 2 | M | 83 | 90 (7) | 98 (8) | 106 (8) |
| 2454_1R | Mouse 3 | M | 86 | 94 (8) | 102 (8) | 110 (8) |
| 2472_1L | Mouse 4 | F | 59 | 66 (7) | 74 (8) | 82 (8) |
| 2473_1R | Mouse 5 | M | 45 | 54 (9) | 62 (8) | 70 (8) |
| 2474_1R1L | Mouse 6 | F | 56 | 65 (9) | 73 (8) | 81 (8) |

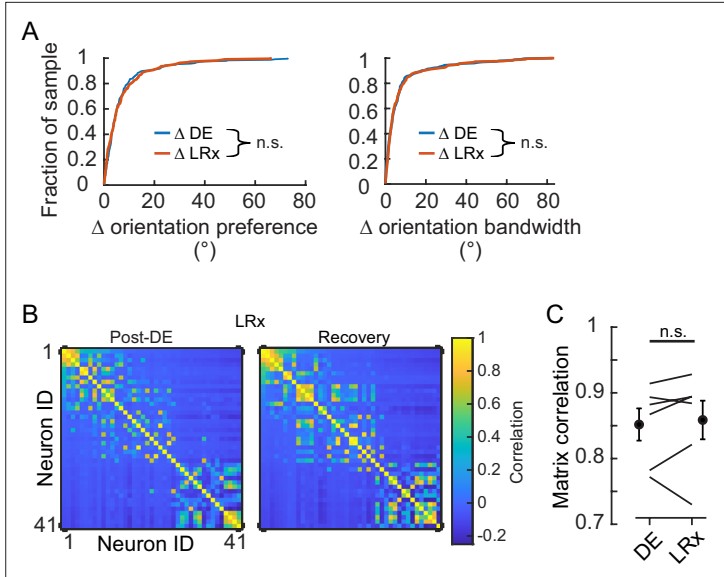

**Figure 2.** Light reintroduction (LRx) did not disrupt tuning in V1 neurons. (**A**) Neither the change in orientation preference or bandwidth was different between dark exposure (DE) and LRx conditions (Wilcoxon rank-sum test corrected for two multiple comparisons [z:–0.3098 and 0.2599, power$_e$: 0.110 and 0.100], p=0.795 and p=0.999, respectively; n=230 and n=216, respectively). All tracked and tuned neurons were included. (**B**) Example of signal correlation matrices from one animal comparing the post-DE (pDE) and recovery (Rec) imaging sessions. For visualization of correlation structure, neuron #1 was randomly selected, and the remaining neurons were sorted in descending order. The same sort-matrix was applied to both sessions. (**C**) Similarity of signal correlation matrices for a given animal (computed as the Pearson's correlation coefficient) was not different across the DE condition compared to the Rec condition (Wilcoxon rank-sum test [*ranksum: 35, power$_e$: 0.999*], p=0.589, n=6 animals). All tracked and tuned neurons were included.

the DE condition when tuning was not part of the inclusion criteria (Wilcoxon rank-sum test, p=0. 015; *Figure 1—figure supplement 1E*). Thus, in contrast to monocular deprivation (*Rose et al., 2016*), a lack of vision through both eyes did not perturb tuning and in fact resulted in a net stabilization.

In contrast to orientation tuning, deprivation-induced changes in s.f. were not readily detected. These results are consistent with prior work indicating that the stability of s.f. preference and orientation preference are independently regulated (*Jeon et al., 2018*). It would be of interest in future work to examine whether at the level of receptive field sub-structure, the axis of receptive field orientation is preferentially destabilized relative to the size of on-off sub-regions. Alternatively, larger sample sizes could reveal subtle changes in s.f.; this would be an indication that instability of s.f. contributes to the observed change in signal correlation.

The above analysis considered the two pools of neurons separately, those tracked and tuned on the B1 and B2 sessions, and those tracked and tuned on the B2 and pDE sessions. Therefore, it is possible that individual neurons themselves were not stabilized by DE, rather, initially unstable neurons became either unresponsive or untuned on the pDE session. To directly determine whether individual neurons became stabilized, we examined that the pool of neurons that were tracked and tuned across all three sessions, B1, B2, and pDE. The average shift in orientation preference across animals was significantly lower in DE compared to the control condition (paired t-test, p=0.029; *Figure 1—figure supplement 1F*). The majority of neurons (58%) was more unstable in the control condition, and the fraction of neurons displaying large shifts (e.g. >50°) was larger in the control condition compared to the DE condition. Consistent with this qualitative description, when pooled across animals, the magnitude of the shift in orientation preference was significantly larger in the control condition compared to the DE condition (paired t-test, p=0.006; *Figure 1—figure supplement 1G*).

Next, we examined whether LRx, which is known to potently induce gene expression even in the adult (*Mardinly et al., 2016*), disturbed tuning stability of the four tuning parameters or pairwise signal correlation. The distribution of the change in orientation preference and bandwidth for the DE condition was indistinguishable from that of the LRx condition (*Figure 2A*). This was also the case

for the other two tuning parameters (median Δ s.f. preference, DE: 5.3E-3 and LRx: 4.9E-3, Wilcoxon rank-sum corrected for four multiple comparisons, p=0.999; median Δ s.f. bandwidth, DE: 5.5E-3 and LRx: 5.2E-3, Wilcoxon rank-sum corrected for four multiple comparisons, p=0.999). Consistent with these results, no difference in the similarity of signal correlation was detected between the DE and LRx conditions (*Figure 2B and C*). Taken together, these data demonstrate that 8 days of DE stabilized tuning response curves to grating stimuli and LRx did not induce a shift in tuning stability when considering those neurons that remained tuned after DE.

## Stimulus representation rebounds within eight days of LRx

Not all neurons are well tuned to grating stimuli, yet such neurons can contribute to visual processing (*Levy et al., 2020*). Therefore, we expanded our analysis to include all tracked neurons, regardless of their responsiveness or tuning characteristics to grating stimuli.

We used a k-nearest neighbor (KNN) classifier to decode stimulus identity, as a measure of neural discriminability. First, we designed the classifiers such that they were trained separately for each session type to estimate the amount of stimulus information contained within the network. The identity of the neurons used was the same for all three sessions, referred to as the 'tracked pool.' Three imaging sessions were examined, the baseline session that immediately preceded DE (B2), the session that immediately followed DE (pDE), and the last session which occurred 8 days after LRx was initiated (Rec). We found that accuracy significantly decreased by 15 ± 3% ( ± SEM, across animals) on the pDE session compared to the B2 session (paired t-test, p=0.046) and recovered within 8 days (B2 versus Rec sessions, paired t-test p=0.16; *Figure 3A*). Visual inspection of the confusion matrices indicated that accuracy was degraded across all spatial frequencies (*Figure 3B and C*). To confirm and quantify this observation, two separate classifiers were constructed for all stimuli less than or equal to 0.1 cycles/° (low s.f.) and all stimuli greater than 0.1 cycles/° (high s.f.). In both cases, accuracy was significantly lower on the pDE imaging session (paired t-test, p=0.037 and p=0.028, respectively; *Figure 3D*).

A possible explanation for the decrease in decoding accuracy is that the response reliability (*Sadeh and Clopath, 2022*) of individual neurons or the number of tuned neurons decreased in the pDE session. Either scenario would be consistent with the observations reported in *Figure 1D*. To determine whether these factors contributed to the decrease in decoding accuracy, first the trial-to-trial response reliability of tuned neurons was compared between the B2 and pDE sessions. Within the tracked pool, a decrease in response reliability for the preferred stimulus was observed. The median reliability of tuned and tracked neurons for individual mice was significantly lower in the pDE session compared to the B2 session (paired-t test, p=6.0E-3; *Figure 3F*). Analysis of within-neuron comparisons corroborated this interpretation and in addition demonstrated that in control conditions, reliability on the initial imaging sessions was predictive of reliability on the subsequent session (*Figure 3—figure supplement 1A, B*). Furthermore, low reliability was associated with loss of tuning on the subsequent session (*Figure 3—figure supplement 1C*).

Similarly, when the reliability of all tuned neurons, regardless of whether tracked, was pooled across mice, the reliability was significantly lower in the pDE session (Wilcoxon rank-sum p=3E-6; *Figure 3G*). It was previously reported that basal firing rates transiently increase when animals are transitioned from prolonged DE to light; such an increase could impact our calculation of reliability. Consistent with the observation that the increase in basal firing rate was prominent within the first 10 min of the dark-to-light transition and returned to baseline within 30–60 min (*Torrado Pacheco et al., 2019*), we did not detect a difference in basal activity between the B2 and pDE imaging sessions (*Figure 3—figure supplement 1D*).

Second, the tracked pool neurons depicted in *Figure 3A–D and F* were scored as being tuned or not tuned across the three sessions. On average across the six mice, there was an 11 ± 4% ( ± SEM) decrease in the fraction of tuned neurons from B2 to pDE. Of the 11% that were lost, approximately 6% were re-gained. We noted that approximately 5% of previously untuned neurons became tuned de novo during the LRx condition, as such the total number of tuned neurons was restored (*Figure 3—figure supplement 1E, F*). Therefore, it is likely that both a decrease in reliability and loss of tuning contributed to lower decoding accuracy in the pDE session.

A potential concern in interpreting these data would be if the pupil was more constricted in the pDE session relative the other sessions. If the pupil diameter was smaller in the pDE session, less light

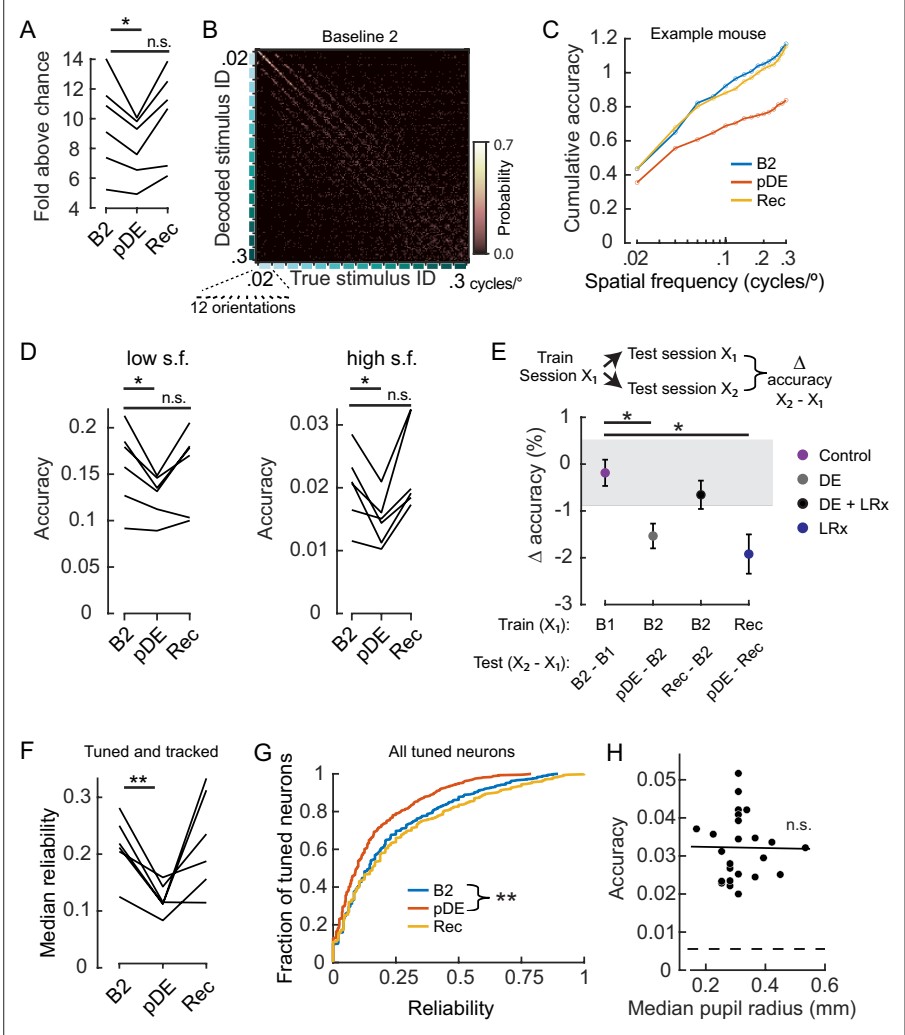

**Figure 3.** Dark exposure transiently decreased neural discriminability. (**A**) Classification accuracy was significantly lower during the post dark exposure (pDE) imaging session compared to the Baseline 2 imaging session (**B2**), and recovered within 8 days of light reintroduction (LRx; paired t-test corrected for two multiple comparisons [*df: 5, t: 3.234*], p=0.046 and 0.16; n=6 animals). All neurons that were tracked across the B2, pDE and Recovery (Rec) sessions were included. See Methods for number of neurons and trials. Chance was 0.0056. (**B**) Example of a confusion matrix from one animal during the B2 imaging session. (**C**) Cumulative sum of classifier accuracy, same mouse as in 'B.' The classification accuracies of the 12 orientations for a given spatial frequency were averaged to produce 15 data points. (**D**) Classifiers using only low (≤0.1 cycles/°, 60 stimuli) and high (>0.1 cycles/°, 120 stimuli) spatial frequency stimuli were decoded separately. In both cases, accuracy decreased during the pDE imaging session compared to the B2 session (paired t-test corrected for two multiple comparisons [*df: 5, t: 3.434 and 3.687*], p=0.037 and p=0.028, respectively), and recovered within 8 days of LRx (paired t-test corrected for two multiple comparisons [*df: 5, t: 0.417 and −1.286, power: 0.063 and 0.188*], p=0.69 and p=0.25, respectively). Chance was 0.0167 and 0.0083, respectively. (**E**) Fixed classifiers were used to quantify representational drift (n=6 animals, the mean ± SEM). The session used to train the classifier is indicated, as well as the specific sessions tested and the condition label. Note, the rate of drift between the Rec and B2 imaging sessions was comparable to ±1 STD of the baseline drift (gray). In contrast, the rate of drift was significantly higher during the DE and LRx conditions compared to the control condition (paired t-test, corrected for three multiple comparisons, values left to right [df: 5,t: 4.056, 1.086, and 3.280], p=0.0293, 0.327, and 0.0329). All neurons that were tracked for a given tested session pair were included. (**F**) The median reliability of responses to the preferred stimulus for a given animal was significantly lower during the pDE imaging session compared to the B2 imaging session (paired t-test corrected for two multiple comparisons [*df: 5, t: 5.474 and −0.2081*], p=0.006 and 0.843); n=6 animals. All tracked and tuned neurons for a given session were included. (**G**) Reliability of individual neuron responses to the preferred stimulus, pooled across animals, was significantly reduced during the pDE imaging session and recovered within 8 days of

*Figure 3 continued on next page*

*Figure 3 continued*

light reintroduction (Wilcoxon rank-sum corrected for two multiple comparisons [*z: 4.670* and *–0.9493*], p=6.0E-06 and 0.343). All tuned neurons were included, n=532 B2, n=419 pDE, and n=450 Rec. (**H**) Decoding accuracy was not correlated with pupil radius (Pearson's correlation [*r: –0.013*], p=0.951). All imaging sessions from six animals were included. Least-squared line indicated. Neuron number (36) was set to the minimum number of tuned neurons among all animals and sessions; trial number (25) was set to the minimum number of trials among all animals and sessions. * p<0.05 and **<0.01.

The online version of this article includes the following figure supplement(s) for figure 3:

**Figure supplement 1.** Extended reliability and basal activity analysis.

may reach the retina and lead to lower input drive. To determine if this was a factor, we examined pupil radius in relation to the imaging sessions. Rather than a decrease, we detected an increase in pupil radius in the pDE session (median radius in millimeters, B1, B2, pDE, and Rec sessions, respectively [ ± SEM]: 0.29±0.01, 0.28±0.02, 0.41±0.03, and 0.29±0.03). However, pupil diameter did not correlate with decoding accuracy (*Figure 3H*), therefore we can rule out the possibility that a lower amount of light reaching the retina accounts for the transient increase in representational drift. We noted that on average the mice ran a little more throughout the pDE session (fraction of session spent in locomotion, B1, B2, pDE, and Rec sessions, respectively [ ± SEM]: 0.18±0.05, 0.10±0.05, 0.27±0.06, and 0.07±0.03). Given that pupil diameter is positively correlated with locomotion (*Reimer et al., 2014*), it is likely that locomotion drove the increase in pupil diameter.

In our final analysis, to assess the rate of representational drift, fixed classifiers were employed, in which a single session was used to train the classifier. That same session ($X_1$), as well as a second session ($X_2$), was used for testing the accuracy of stimulus identify classification. The rate of drift was defined as the difference in accuracy between $X_2$ and $X_1$. The rate of drift was significantly higher during DE (sessions pDE and B2) compared to the control condition (sessions B2 and B1; paired t-test, p=9E-3; *Figure 3E*). These results are an indication that similar to the olfactory cortex (*Schoonover et al., 2021*), continuous sensory experience reduces representational drift.

Notably, there was not a difference in the change in decoding accuracy across the control condition and between the Rec and the B2 sessions, even though the time span covered was 16 days (DE +LRx) rather than 8 days. These results demonstrate that the stimulus representation not only recovered after transient DE but that the representation rebounded to its original state. To confirm this interpretation, fixed classifiers were trained using the neural activity from the Rec session, and the difference in accuracy between the pDE (the 'second' test session in this case) and Rec sessions was computed. As expected, the rate of drift was higher than in the control condition (paired t-test, p=0.012; *Figure 3E*).

In summary, we found that in the adult, stimulus encoding is robust to transient deprivation and is capable of recovering not only in terms of the estimated amount of stimulus information contained in V1 but also that the stimulus representation rebounds to its original form. LRx, despite initiating a cascade of changes in gene expression (*Torrado Pacheco et al., 2019*; *Mardinly et al., 2016*), did not persistently disturb stimulus encoding. Thus, using DE as a treatment for amblyopia is not expected to have a negative impact on previously established visual function. Furthermore, our results establish that exposure to naturalistic statistics in the home-cage environment improves neural discriminability well into adulthood. It will be of interest in future studies to determine whether stimulus discrimination reaches a plateau and requires continuous experience to maintain the plateau or continues to improve past the classic critical period for ocular dominance plasticity.

## Ideas and speculation

Our DE experiments revealed that vision has a destabilizing influence on the persistence of orientation preference for approximately 30% of the population of imaged neurons. Similarly, the persistence of pairwise signal correlation among V1 neurons was lower in control conditions compared to the DE condition. Thus, it is possible that under normal natural viewing conditions in the adult, orientation preference is subjected to on-going Hebbian plasticity which has an observable net destabilizing effect on the preference of individual neurons in V1; these changes potentially occupy the null coding space such that they do not impact downstream readout in the visual hierarchy but are sufficient and necessary to maintain retinotopic and matched organization of receptive field structure. In this

scenario, DE-induced stabilization would not be a direct consequence of mechanisms associated with deprivation-induced homeostasis (*Bridi et al., 2018*; *Turrigiano, 2008*) rather the removal of a disrupting influence.

The observations reported here are consistent with the theoretical proposal that there is a plastic substrate of neurons with preferentially higher recurrent connectivity that coexist with a stable 'backbone' formed by neurons that are resistant to sensory perturbations (*Sweeney and Clopath, 2020*). Such a functional architecture has the advantage that new information can be integrated into existing networks without perturbing on-going function and could contribute to stable perception while allowing for adaptive flexibility. In other words, the backbone is resistant to effects of homeostatic plasticity induced by DE. Also, consistent with the proposed functional architecture, we found that that the decoding accuracy of fixed classifiers rebounded to their original, pre-dark exposed state during LRx. The extent to which a stable backbone could be updated to integrate previously deprived input in the case of pathologies such as amblyopia is unclear. Prior work demonstrates that visual perceptual training is effective at improving spatial acuity specifically if the training immediately follows DE (*Eaton et al., 2016*) such that the rejuvenating effects of DE are still present. It would be of interest in future experiments to determine whether DE transiently allows for a stable backbone to be updated to reflect newly integrated input.

Notably, the development of natural scene processing is protracted relative to grating stimulus encoding in V1 (*Kowalewski et al., 2021*). Specifically, a proportion of V1 neurons develops a strong preference for complex scenes relative to simple grating stimuli (*de Vries et al., 2020*; *Kowalewski et al., 2021*; *Walker et al., 2020*). It will be of future interest to determine whether the development of complex scene processing is facilitated by the presence of neurons that are functionally distinguishable from a stable 'backbone' and the extent to which complex-scene preferring neurons emerge from a potentially more plastic population within V1. Such an architecture may be reinforced by a spatial segregation of neuomodulatory input arising from subcortical brain regions associated with reward signals, such as the nucleus basalis (*Pafundo et al., 2016*; *Hangya et al., 2015*; *Ji et al., 2015*). In this scenario, a stable 'backbone' could serve the purpose of retaining spatial receptive field position and binocular alignment (*Wang et al., 2010*; *Sarnaik et al., 2014*; *Chang et al., 2020*) between the two eyes during the protracted development of complex scene processing.

## Methods
### Animal preparation and timeline of imaging

All experimental procedures were compliant with the guidelines established by the Institutional Animal Care and Use Committee of Carnegie Mellon University and the National Institutes of Health, and all experimental protocols were approved by the Institutional Animal Care and Use Committee of Carnegie Mellon University (protocol # PROTO201600014). To express the calcium indicator GCaMP6f selectively in excitatory neurons, homozygous Emx1cre mice (Jackson Laboratories, stock number 005628) were crossed with homozygous Ai93/heterozygous Camk2a-tTA mice (Jackson Laboratories, stock number 024108). Experimental mice were heterozygous for all three alleles. Mice were housed in groups of 2–3 per cage, in a 12 hr light/12 hr dark cycle; all imaging sessions started at Zeitgeber time (ZT) 14.5±1, where ZT0 is lights on, and ZT12 is lights off. The same enrichment materials were provided in all cages including a Plexiglas hut and nesting material. Mice were housed in the barrier section of the animal facility until the time of surgery, luminance in this room was 240 lux. After surgery mice were maintained in a different location that also had a luminance of 240 lux. During animal set up and tracking prior to recording luminance was 60 lux. See *Table 1* for information on animal sex and genotype. Males and females were randomly selected. None of the mice used in this study exhibited aberrant, interictal events (*Kowalewski et al., 2021*; *Steinmetz et al., 2017*) in V1 or adjacent regions.

Mice (29–31 days old) were anesthetized with isoflurane (3% induction, 1–2% maintenance). A 3 mm diameter craniotomy was made over the primary visual cortex in the left hemisphere, identified by coordinates and landmarks as described in *Feese et al., 2018*. A stainless-steel bar, used to immobilize the head for recordings, was glued to the right side of the skull and secured with dental cement. The craniotomy was then covered with a double glass assembly in which the diameter of the inner glass was fitted to the craniotomy and sealed with dental cement. To ensure correct targeting

of recordings to the binocular zone of V1, in a pilot experiment, visual stimuli were presented to the ipsilateral pathway. The screen was placed directly in front of the animal, perpendicular to the midline. The contralateral eye was occluded by attaching an opaque shield to the microscope objective. Mice typically adapt to this irritant within 1–2 min. We confirmed that our imaging field of view could be placed such that the field of view did not extend past the boarders of binocular V1 (*Figure 1— figure supplement 2*). In the pilot experiment, the calcium indicator GCaMP6f was expressed via a virus (Addgene #100837: pAAV.Syn.GCaMP6f.WPRE.SV40; titer $\geq 1 \times 10^{13}$ vg/mL). Two injections were targeted to the binocular zone of V1 in the left hemisphere (0.3 mm anterior to lambda; 3 and 3.25 mm lateral to the midline, respectively) using a glass micropipette and a PicoSprizer III Microinjector (20 psi, 10–80 ms pulses, 2 s pulse interval). Injections started at an initial depth of 400 μm. The pipette was then raised in 25 μm steps, and equal amounts of virus were injected up to a final depth of 100 μm below the dura for a total injected volume of ~300–500 nl across layer 2/3.

Prior to the first baseline recording session in the experimental mice, the location of binocular neurons in V1 was identified using the following search strategy: neural activity in response to ipsilateral pathway stimulation was visualized at low magnification using a field of view sized approximately 900×900 microns and moved until the location of the highest-intensity ipsilateral-driven activity was identified. This location was centered in the imaging field of view and the same location used in subsequent imaging sessions. The search area was restricted to an area 2.7 mm posterior of Bregma, thus ensuring extra striate regions anterior to V1 were avoided. Note, although the lateral medial area adjacent to the binocular zone contains binocular neurons (*Wagor et al., 1980*), the responses are weaker compared to V1 (*Kalatsky and Stryker, 2003*). After the initial screening described above, four imaging sessions were acquired, each session was separated by 8±1 days (see *Table 1* for the precise age of each animal on each imaging session). Animals were dark-exposed for 8 days. On the pDE imaging session, animals were removed from the dark and exposed to low-light conditions (60 lux) for 30–40 min during the animal mounting and neuron tracking procedure, prior to data collection.

## Data acquisition, neuron segmentation, and neuron tracking

Two-photon calcium imaging was performed in awake head-fixed mice mounted atop a floating spherical treadmill using a resonant scanning microscope (Neurolabware) outfitted with a 16× Nikon objective (0.80 NA) and 8 kHz resonant scanning mirror. Treadmill motion was recorded using a camera (Dalsa Genie M640-1/3) for off-line analysis of locomotion (*Jeon et al., 2018*), and eye blinks were captured using a second camera (Dalsa Genie M1280; *Kowalewski et al., 2021*). A laser excitation wavelength of 920 nm was used (Coherent, Inc); green emissions were filtered (Semrock 510/84–50), amplified (Edmund Optics 59–179), and detected with a photomultiplier tube (PMT; Hamamatsu H1 0770B-40). The imaged field of view was 620×504 microns, pixel dimensions were 0.85×0.98 μm, and the acquisition rate was 15.5 Hz. The acquired image time series were motion-corrected by computing the horizontal and vertical translation of each frame using phase correlation (*Kowalewski et al., 2021*) and individual neurons segmented using the Matlab version of Suite2p toolbox (*Pachitariu et al., 2017*), as described in *Kowalewski et al., 2021*.

To identify neurons that were tracked across imaging sessions, we registered repeat imaging sessions using the mean intensity image of each session. The mean intensity image for a session was computed by averaging the intensity of each pixel in the aligned calcium image series across time for the entire imaging session (roughly 50,000 frames). Then, the mean intensity images of the two sessions were registered using an affine transform with one-plus-one evolutionary optimizer. Once the sessions were registered, the percentage of pixel overlap between the neurons from two sessions was computed. Neurons were accepted to be the same neuron across sessions if the percentage of overlapping pixels across the two sessions was larger than 75%. On average, there were 160 pixels in a given neuron.

## Visual stimulation

Static sinusoidal grating stimuli were generated using psychophysics toolbox (http://psychtoolbox. org/) in Matlab (Mathworks, Boston, MA, USA). Stimuli were presented at 100% contrast; the luminance output of the screen was 17 cd/m$^2$. The stimulus was presented on a screen positioned 25 cm away from the right eye angled at 50° with respect to the midline of the animal. The size of the screen

was 64×40 cm, thereby subtending 142×96° of visual angle. The s.f. range of the stimulus set was 0.02 cycles/° to 0.3 cycles/° at 0.02 cycles/° interval. The orientations ranged from 0 to 180° at 15° spacing interval, yielding a total of 180 different sinusoidal gratings with 12 different orientations and 15 different spatial frequencies. Each grating was presented for 250 ms consecutively in a random order without interleaved gray screen; this was repeated four times, and data were saved to disk. This sequence was repeated a minimum of nine times, resulting in a total of at least 36 trials for a given stimulus. Taking into account trials removed due to locomotion or pupil tracking (see Quantification of visual responses), a minimum of 25 trials was used in analysis. 2 s of isolumant gray screen was presented at the onset of each sequence.

## Quantification of visual responses

Reverse correlation was used to determine the response window of a given stimulus (**Ringach et al., 1997**). The peak in the stimulus-averaged events was observed 194–320 ms after the stimulus was presented on the screen. Therefore, for each stimulus, the corresponding event activity was computed by averaging the number of events between 194 ms and 320 ms window. We defined this period as the response window for a given stimulus.

A neuron was defined as responsive to visual stimuli when the number of events following a presentation of a visual stimulus was modulated by the stimuli presented. To test for modulation, we performed a one-way ANOVA($\alpha$=0.01) on the observed events during the response window using stimuli as the factor for each neuron.

GCaMP6f expressed in neurons has a longer decay than the presentation rate of our stimuli (**Chen et al., 2013**). Therefore, we used deconvolution to remove the effects of decay in calcium fluorescence in quantifying responses of each neuron to our visual stimuli as in **Jeon et al., 2018**. The amplitude of calcium transients was expressed in units of inferred events. For each segment n, inferred events $s_n$ were estimated from fluorescence using the following model:

$$f_n = s_n * k + \beta_n p_n + b_n$$

where k is the temporal kernel, and $b_n$ is the baseline fluorescence. Neuropil fluorescence, which is a contamination of the fluorescence signal $f_n$ from out of focus cell bodies and nearby axons and dendrites, is modeled by $p_n$, the time course of the neuropil contamination, and, $\beta_n$ the scaling coefficients * denotes convolution. Using this model, $s_n$, k, $\beta_n$, and $b_n$ were estimated by a matching pursuit algorithm with L0 constraint, in which spikes were iteratively added and refined until the threshold determined by the variance of the signal was met.

Trials containing locomotion or eye blinks were removed. Pupil location was estimated from eye-tracking videos using a circular Hough transform algorithm; the algorithm failed to find the pupil on frames during which the mice were blinking. These frames were marked as eye blink frames and removed from further analysis. Trials with locomotion were identified as in **Jeon et al., 2018**. Briefly, after applying a threshold on the luminance intensity of the treadmill motion images, phase correlation was computed between consecutive frames to estimate the translation between the frames. To define a motion threshold, the data were smoothed using a 1 s sliding window. Any continuous non-zero movement periods during which the animal's instantaneous running speed exceeded 10 cm/s threshold for at least one frame were marked as running epochs.

## Estimation of preferred stimulus and tuning bandwidth

Orientation and s.f. preference were determined using a two-dimensional Gaussian model, fit to single trial responses. For neurons that were responsive to grating stimuli, a two-dimensional Gaussian model was fit using non-linear least-squared regression such that the number of events R as a function of the orientation θ and the s.f. φ of the stimulus was

$$R\left(\theta, \varphi\right) = \frac{A}{2\pi\sigma_\theta\sigma_\varphi\sqrt{1-\rho^2}} e^{\left(-\frac{1}{2(1-\rho^2)}\left[\frac{(\theta-\mu_\theta)^2}{\sigma_\theta^2} + \frac{(\varphi-\mu_\varphi)^2}{\sigma_\varphi^2} - \frac{2\rho(\theta-\mu_\theta)(\varphi-\mu_\varphi)}{\sigma_\theta\sigma_\varphi}\right]\right)} + B$$

where $\mu_\theta$ was the preferred orientation, and $\mu_\varphi$ was the preferred s.f. of the stimulus, and the $\sigma_\theta$ and $\sigma_\varphi$ described the widths of respective tuning. The covariance of responses for orientation and s.f. was captured by the correlation term $\rho$. A was a parameter accounting for the amplitude of the responses

in number of events, while B was the baseline event activity of the cell. For fitting, the lower and the upper bound of allowed values for $\mu_\varphi$ were set by the range of the presented stimuli, which was 0.02–0.30 cycles/°. The lower bound for $\sigma_\theta$ and $\sigma_\varphi$ was set at 1° and 0.001 cycles/°, respectively, to prevent fits with zero or negative widths. Prior to fitting, the preferred orientation was initialized by estimating the preferred orientation by averaging the response, $R$ across all spatial frequencies for a given stimulus orientation, $\theta$ and calculating half the complex phase of the value (*Niell and Stryker, 2008*; *Kuhlman et al., 2011*).

$$S = \frac{\sum R(\theta)e^{2i\theta}}{\sum R(\theta)}$$

The preferred s.f. was initialized by selecting the s.f. that generated the maximal significant response at the estimated preferred orientation. For the model above, the goodness of fit ($R^2$) was used to identify neurons with significant tuning. The chance distribution of $R^2$ values was calculated from fitting the above model with permuted stimulus labels on individual trials 1000 times for each neuron. Neurons whose $R^2$ exceeded the 95th percentile of the chance $R^2$ distribution were accepted as significant and referred to as well tuned to grating stimuli. Fitting on individual trials ensures that only neurons with responses that are similar across trials are considered well tuned. The tuning curves of all such neurons are provided in an associated metadata file, '*Source data 1*.' Note, the goodness of fit $R^2$ values is calculated using individual trials, as such are lower than what is typically observed for fits using trial-averaged responses. The associated metadata demonstrates that the fits are an accurate representation of the data.

The bandwidths of the Gaussian tuning were described using half-width at half-maximum (HWHM). The HWHM bandwidths for both orientation and s.f. were calculated as

$$\mathrm{BW} = \sqrt{2 * \ln(2)} * \sigma$$

where $\sigma$ was the width parameter of the Gaussian fit.

## Computation of signal correlation (*Figure 1F, G*; *Figure 2B*)

Signal correlation $\rho^{sig}$ between a pair of neurons is defined as Pearson's correlation between the average responses to stimuli (*Averbeck et al., 2006*). Therefore, we computed pairwise signal correlation between neuron i and neuron j as

$$\rho_{i,j}^{sig} = corr(\bar{R}_i, \bar{R}_j)$$

where $\bar{R}$ is a vector of average response in number of spikes to 180 sinusoidal gratings for the respective neuron.

## Stimulus classification (*Figure 3A-E*)

KNN classifiers were used to decode the presented stimuli from vectors of single trial population responses (*de Vries et al., 2020*; *Kowalewski et al., 2021*). In our case, the KNN classifier estimated the stimulus identity for a given response vector by identifying the most frequent stimulus identity of its k closest response vectors. To identify the nearest neighbors for a given response vector, we computed the Euclidean distance to the other response vectors. For each session, data were divided so that a single set of response vectors consisted of one trial of each stimulus. This resulted in the number of sets being equal to the number of trials that each stimulus was shown. When the number of trials available was larger than the minimum number of trials, trials were randomly subsampled from the available trials. During decoding, the possible neighbors for a test response vector consisted of all response vectors not belonging to the test set. This ensures an unbiased representation of possible nearest neighbors across stimuli. This process was repeated across each response vector and each set. We reported the performance of this decoding process as accuracy across all response vector tested. Previously, we found that a value of k=4 resulted in the best average rank across mice (*Jeon et al., 2021*), therefore we fixed the value of k to 4. Chance performance of the classifiers was 1 divided by the number of stimuli classified. The number of trials and neurons was matched to the minimum number of available trials and neurons across the three sessions, B2, pDE, and Rec. In the case, more trials or neurons were available; neurons and trials were randomly subsampled.

To quantify representation drift, we modified the KNN classifier described above. We trained a KNN classifier on the neuronal responses from a single session and decoded held-out responses from that given session and as well as responses from a second session acquired on a different day. Only the neurons that were tracked in both sessions were used to train and test the classification algorithm. The number of trials was matched to the minimum number of available trials across the two sessions.

The number of neurons and trials used were as follows: *Figure 3A and D*, for mouse # 1–6, the number of trials was 42, 31, 33, 27, 26, and 37 for each of the imaging sessions, and the number of neurons was 114, 98, 75, 64, 55, and 96, respectively; *Figure 3E*, the number of trials ranged between 25 and 44 and the number of neurons ranged between 68 and 167, depending on the animal and condition. *Figure 3H*, the number of trials was 25, and the number of neurons was 36.

### Reliability across trials (*Figure 3F, G*)

Reliability was computed as the proportion of trials in which the response amplitude was at least 2 SDs above baseline activity, where baseline activity was defined as the activity during presentation of the isolumant gray screen.

## Two-sample proportions Z-test (Figure 1—figure supplement 2C)

We computed the Z-statistic for the hypothesis that the fraction of untuned neurons came from two separate binomial distributions against the null hypothesis that they came from the same distribution. The Z-statistic for the difference in performance between the first session and the $i^{th}$ session was computed by the following equation.

$$Z = \frac{p_1 - p_i}{\text{SE}}$$

SE was the standard error of the sampling distribution difference between the two performances and $p_1$ and $p_i$ were the performance of the first session and the $i^{th}$ session, respectively. SE of the first session and the $i^{th}$ session was computed by the equation below,

$$\text{SE} = \sqrt{p * (1 - p) * \left(\frac{1}{n_1} + \frac{1}{n_i}\right)}$$

where $p$ is the pooled performance between session 1 and session $i$, weighted by the number of trials, $n$, of the respective session.

$$p = \frac{p_1 * n_1 + p_i * n_i}{n_1 + n_i}$$

p Values were computed from the Z-statistic.

## Statistics

Error is reported as SEM, unless noted. In the case data were not normally distributed, non-parametric tests were used. Alpha was set to 0.05 unless noted. In the case p values were corrected for multiple comparisons, the number of comparisons is noted in the figure legend; correction was computed using the false discovery rate Benjamini-Hochberg procedure. Degrees of freedom (df) and the test statistic are reported in italics in the figure legends. In the case of small sample sizes when using the Wilcoxon rank-sum test, the rank-sum statistic is reported when the approximate method is not used. The fraction of neurons (normalized to the total number of neurons segmented with a given imaging sessions) that were responsive is reported in *Supplementary file 1A*. Furthermore, the fraction of neurons that were included in each figure panel is listed in *Supplementary file 1B-D*; in the case the SEM was non-overlapping between the first imaging session and a given imaging sessions, the SD is included and paired t-test p values, adjusted for multiple comparisons as appropriate, are indicated. Prior to initiating the study, based on prior work (*Jeon et al., 2021*), it was estimated that a sample size of n=6 animals would be sufficient to detect differences if present. Reported power was computed using IBM SPSS in the case of t-tests, and a power estimate ($power_e$) was computed in the case of Wilcoxon rank-sum tests using the Matlab function 'sampsizepwr' with alpha set to 0.01.

## Materials availability statement

No new biological materials were created in this study.

## Code availability

The code used for analysis is available on GitHub (https://github.com/bjjeon5111/BCI_V1, copy archived at swh:1:rev:e026800170840d001bd27632db2a2008798653a4; *Jeon, 2022*).

## Acknowledgements

We thank Jeffrey Good for performing surgeries. Funded by: NIH R01EY024678 (SJK) and The Curci Foundation (SJK and SMC).

## Additional information

### Funding

| Funder | Grant reference number | Author |
|---|---|---|
| National Institutes of Health NEI | R01EY024678 | Sandra J Kuhlman |
| Curci Foundation | | Sandra J Kuhlman Steven M Chase |

The funders had no role in study design, data collection and interpretation, or the decision to submit the work for publication.

### Author contributions

Brian B Jeon, Conceptualization, Data curation, Formal analysis, Investigation, Methodology, Writing – original draft, Writing – review and editing; Thomas Fuchs, Data curation, Investigation; Steven M Chase, Formal analysis; Sandra J Kuhlman, Conceptualization, Resources, Formal analysis, Supervision, Funding acquisition, Validation, Methodology, Writing – original draft, Project administration, Writing – review and editing

### Author ORCIDs

Sandra J Kuhlman http://orcid.org/0000-0003-0450-7282

### Ethics

All experimental procedures were compliant with the guidelines established by the Institutional Animal Care and Use Committee of Carnegie Mellon University and the National Institutes of Health, and all experimental protocols were approved by the Institutional Animal Care and Use Committee of Carnegie Mellon University (protocol # PROTO201600014) .

### Decision letter and Author response

Decision letter https://doi.org/10.7554/eLife.80361.sa1
Author response https://doi.org/10.7554/eLife.80361.sa2

## Additional files

### Supplementary files

• Supplementary file 1. Fraction of neurons included, organized by figure. (A) Fraction of responsive neurons: Data are normalized to the number of neurons segmented in a given session. The SEM was non-overlapping between Baseline 1 (B1) and post dark exposure (pDE), therefore one STD is included. The fraction of responsive neurons in B1 versus pDE was not significantly different (paired t-test [*df: 5, t: 2.213, power: 0.433*], p=0.078; n=6 animals). (B) Fraction of neurons included in *Figures 1 and 2*: Data are normalized to the number of neurons segmented in a given session. The SEM was overlapping in all cases. (C) Fraction of neurons included in *Figure 3*: Data are normalized to the number of neurons segmented in a given session. In *Figure 3A and D*, the SEM was non-overlapping between Baseline 2 (B2), pDE, and Rec, therefore one STD is included; the fraction of included neurons in B2 versus pDE and B2 versus Rec was not significantly different (paired t-test [*df: 5, t: −2.458,–2.685, power: 0.509, 0.580*], adjusted for two multiple comparisons, p=0.0574 and p=0.0871, respectively; n=6 animals). In *Figure 3E*, the SEM was non-overlapping between B1

and B2 (pDE-B2), therefore one STD is included; the fraction of included neurons in B1 versus B2 (pDE-B2) was not significantly different (paired t-test [*df: 5, t: 2.50, power: 0.523*], p=0.0545; n=6 animals). (D) Fraction of neurons included in *Figure 1—figure supplement 1*: Data are normalized to the number of neurons segmented in a given session. The SEM was overlapping in all cases.

• MDAR checklist

• Source data 1. Two-dimensional Gaussian tuning curve fits. Tuning curves for all neurons scored as significantly tuned and tracked, for each of the three conditions: control, dark exposure (DE), and light reintroduction (LRx).

## Data availability

The data set analysed in this manuscript is available at the GIN repository, https://doi.org/10.12751/g-node.n8mnh8.

The following dataset was generated:

| Author(s) | Year | Dataset title | Dataset URL | Database and Identifier |
| --- | --- | --- | --- | --- |
| Jeon B, Kuhlman S | 2022 | Dark exposure experiment data | https://doi.org/10.12751/g-node.n8mnh8 | G-Node, 10.12751/g-node.n8mnh8 |

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
