## [Editor Report]

The present manuscript examines cortical representations of basic visual attributes following a manipulation shown to enhance plasticity in the adult brain: binocular dark exposure for 8 days, followed by light re-introduction. Prior work did not rule out the possibility that prolonged dark exposure could negatively impact visual representations in V1. Using 2P calcium imaging in awake adult mice to quantify changes in stimulus selectivity, discriminability, and reliability of V1 neurons, Jeon and colleagues provide compelling evidence that dark exposure has opposing but transient effects at the single neuron versus population level, thus failing to permanently disrupt visual representations in V1.

---

## [Decision Letter]

**Decision letter after peer review:**

Thank you for submitting your article "Visual experience has opposing influences on the quality of stimulus representation in adult primary visual cortex" for consideration by *eLife*. Your article has been reviewed by 3 peer reviewers, one of whom is a member of our Board of Reviewing Editors, and the evaluation has been overseen by Joshua Gold as the Senior Editor. The following individual involved in review of your submission has agreed to reveal their identity: Arbora Resulaj (Reviewer #3).

The reviewers have discussed their reviews with one another, and the Reviewing Editor has drafted this to help you prepare a revised submission. Included here is a brief summary and list of essential revisions the reviewers and review editor deem necessary for you to address.

Essential revisions:

As you will be able to read below, reviewers appreciated the study and manuscript, which provides a unique and unexpected set of results for the field of adult visual plasticity, at the levels of both population and single cell responses in rodent V1. The writing was clear, the figures appropriate and importantly, the study design and analyses were deemed rigorous and generally appropriate. However, reviewers raised concerns with regard to some of the claims and data interpretation. Rather than additional experiments, they suggested additional information and analyses of the data already collected. The key points that need to be addressed can be summarized as follows:

1. Several experimental parameters are missing or difficult to locate, and should be provided clearly and completely in the Methods/Results:

a. A clear experimental timeline.

b. Mouse age and its impact/lack of impact on results.

c. Cage environment parameters.

d. Stimulus and screen parameters for behavioral assays.

e. Imaging/recording locations within V1 (monocular versus binocular zone?).

2. Reviewers raised several questions about the proportion (and properties) of untuned neurons in Figure 1 and loss of tuning in Figure 3, as well as change in tuning stability for spatial frequency (as opposed to orientation).

3. Proper statistical reporting, sample size justification and alternative statistical tests are needed to address strength of findings with regard to increased stability of orientation preferences and generalizability of results to the level of single animals.

4. Expanded discussion of results in view of: (1) the work of Schoonove et al., 2021 in olfactory cortex (esp. the impact of repeated stimulus presentations), (2) the role of decreased signal to noise ratio (and/or firing rates) in the observed changes in neuronal tuning after DE, and (3) how the notion that in the intact visual system, neural circuits that are easily perturbed by sensory fluctuations co-exist with more stable neurons that are resistant to them applies to abnormally-formed visual systems – such as those in amblyopes.*Reviewer #1 (Recommendations for the authors):*

There are a number of issues I would like to see addressed in this otherwise well-framed and clearly written manuscript:

Methodological:

– State the length of DE in the Methods and at the first mention of DE in Results. Currently, it is stated in the abstract, then on line 173 of the Results and Figure 1 legend.

– Please provide information about cage environment luminance, as well as visual stimulus screen parameters such as luminance and luminance contrast of the stimuli utilized. Was the screen luminance calibrated for instance? Were all gratings high-contrast? I assume they were – but what contrast?

– 6 mice appears to be a low sample size – as such, there is a question as to whether sufficient power is present for some of the analyses performed. Please address. Related to this, please provide full statistical expression (degrees of freedom, t or F values as appropriate) every time stats are detailed in the text and figure legends.

– Please indicate where in V1, imaging/recordings were done. Was it in the monocular section? The binocular section? Some whole brain reconstructions of the location of imaging sites would be useful in this context. Would you expect to see different effects in monocular versus binocular cells?

– Please provide some evidence (anatomical or otherwise) that you were analyzing neurons in V1 and not extrastriate cortex.

Interpretational:

– While the single parameter (ori preference, ori bandwidth, SF preference, SF bandwidth) analysis reveals a change in tuning stability for orientation after DE, there is none for SF – please discuss why and what the functional implications might be.

– Please discuss the question of why you were able to elicit relevant visual responses from only a small fraction of V1 neurons imaged for the comparisons made between B1-B2, B2-pDE, etc. What exactly caused this (loss of visual responsiveness, or ability to be fit with the 2D Gaussian model)? How does that impact interpretation of the effects of DE?

– The last sentence of the Introduction (lines 104-105) comes out of no-where, stating that "exposure to natural image statistics in the home-cage improves neural discriminability in the adult". It is not clear how the authors arrive at this conclusion, especially here in the Introduction (before methods and results are detailed) and in view of the fact that there is next to no information about the visual environment of their experimental animals in their cages. Were natural image statistics measured? Manipulated? Have the authors established a causal relationship between statistics of natural cage environment and neural tuning or discriminability in their animals or is this speculation? Please clarify. What may additionally help the reader in this context, is for authors to introduce the notion of the natural visual environment earlier, as they describe their experimental manipulations. Light re-introduction is in fact not just re-introduction of light, but re-introduction to a high luminance state + the natural [cage] visual environment.

– Please discuss the temporal requirements for the phenomena observed – or at the very least justify why 8 days were selected for both the DE and the light re-exposure. Would 1 day have sufficed? Would 3? This ties in with my next point ….

– It would be good to see the discussion of results include some mechanistic hypotheses about the phenomena described. Do the authors think the representational stabilization observed is the result of homeostatic plasticity induced by deprivation or a Hebbian mechanism? What about the effect of visual re-exposure?

– Finally, the authors nicely present their work in the context of the theory proposed by Clopath and colleagues, about the co-existence of neural populations that are more easily perturbed by sensory stimuli and those which are more resistant – the more "stable backbone". While the present work may have confirmed such a scenario in the intact mouse visual cortex, it would be useful to see a discussion of the likelihood that this may also be true in pathological visual systems such as those of amblyopic subjects.

*Reviewer #2 (Recommendations for the authors):*

Despite the significance of the work and the findings, there are a number of points that should be clarified in the manuscript:

1. According to supplemental data file 1, 6 mice were recorded in these experiments, 3 aged postnatal day 45-59 and 3 aged postnatal day 83-86. It would be interesting to see if the response to dark exposure was different across the 2 age groups, as predicted by previous work. An experimental timeline would be useful, the above age at imaging initiation is only available in supplemental data.

2. The representational drift demonstrated in Deitch et al., in visual cortex is not stabilized by the visual stimulation protocol, however Schoonove et al., 2021 show demonstrate that daily exposure to the same odorant stabilizes stimulus selectivity in the olfactory cortex. I'd like to see the authors expand their discussion of these interesting observation.

(It would also be more precise for the authors to refer to Schoonove et al., as working in olfactory cortex not association cortex).

3. In Figure 1B, "tracked and tuned" neurons were shown in red and "segmented" neurons were shown in yellow. How about "tracked but not tuned" neurons? Were they included in "segmented" group? Since not all visually responsive neurons are tuned, this group should be separated by "segmented".

4. In Figure 3F and G, the nature of decrease in tuning should be further discussed. Pacheco et al., 2019 demonstrated, prolonged exposure to darkness increases the firing rates upon LRx. Is the decrease in reliability due to the decrease in SNR? Any difference in firing rate changes between "neurons remained tuned" vs "neurons lost tuning"?

5. Unless those 30% of neurons that lost tuning after pDE are the ones regain tuning, visual representation after 8 d LRx is different from pre-DE condition (B1 and B2). It is not clear if the identity of 30% of neurons were tracked after 8 d LRx to prove this.*Reviewer #3 (Recommendations for the authors):*

The paper uses a combination of analysis methods that are both thoughtful and complementary. My main recommendations are for additional analyses to strengthen the findings.

(1) Please mention whether the monocular zone of V1 or the binocular zone of V1 was imaged. This is relevant for comparison with monocular deprivation studies.

(2) The finding that a fraction of neurons show more stable orientation preferences following dark exposure depends on the quality of the fits for the tuning curves. While the authors present a control where they remove 10% of neurons with the worst R2 from their analysis, and show that the conclusions still hold, other complementary analysis can be used. Does R2 depend on change in orientation preference? If neurons were to be ranked according to their change in orientation preference and split into three equal groups, is there a difference in R2 between the three groups?

(3) The paper pools visual responses across six mice to show a smaller change in orientation preference following dark exposure compared to baseline. Is there an effect if you do Wilcoxon rank sum test on visual responses from each individual mouse (ie. Figure 1D statistical test performed on visual responses from individual mice)?

(4) Additional analysis is needed to support the following sentence in the Abstract or this sentence should be softened: "a decrease in response reliability across a broad range of spatial frequencies accounted for the disruption". The paper shows that there is both decreased reliability and degraded classifier performance. However, to address whether decreased reliability indeed degrades classifier performance, the authors could remove the least reliable neurons and compare classifier performance (the comparison may have to be with a matched number of neurons).

---

## [Author Response]

Reviewer #1 (Recommendations for the authors):There are a number of issues I would like to see addressed in this otherwise well-framed and clearly written manuscript:Methodological:– State the length of DE in the Methods and at the first mention of DE in Results. Currently, it is stated in the abstract, then on line 173 of the Results and Figure 1 legend.

Updated as recommended.

– Please provide information about cage environment luminance, as well as visual stimulus screen parameters such as luminance and luminance contrast of the stimuli utilized. Was the screen luminance calibrated for instance? Were all gratings high-contrast? I assume they were – but what contrast?

Grating stimuli were 100% contrast, the luminance output of the screen was 17 cd/m^2^. Mice were housed in the barrier section of the animal facility until the time of surgery, luminance in this room was 240 lux. After surgery mice were maintained in a different location that also had a luminance of 240 lux. During animal set up and tracking prior to recording luminance was 60 lux. Methods are updated to include this information.

– 6 mice appears to be a low sample size – as such, there is a question as to whether sufficient power is present for some of the analyses performed. Please address. Related to this, please provide full statistical expression (degrees of freedom, t or F values as appropriate) every time stats are detailed in the text and figure legends.

Power analysis is reported for comparisons in which a difference was not detected. Degrees of freedom and the test statistic are now included, as appropriate.

– Please indicate where in V1, imaging/recordings were done. Was it in the monocular section? The binocular section? Some whole brain reconstructions of the location of imaging sites would be useful in this context. Would you expect to see different effects in monocular versus binocular cells?

Imaging was done in the binocular zone, methods were updated and a supplemental figure panel was added demonstrating how the binocular zone in V1 was identified (Figure 1 —figure supplement 2).

– Please provide some evidence (anatomical or otherwise) that you were analyzing neurons in V1 and not extrastriate cortex.

This information is now provided in Figure 1 —figure supplement 2.

Interpretational:– While the single parameter (ori preference, ori bandwidth, SF preference, SF bandwidth) analysis reveals a change in tuning stability for orientation after DE, there is none for SF – please discuss why and what the functional implications might be.

We added the following: In contrast to orientation tuning, deprivation-induced changes in spatial frequency were not readily detected. These results are consistent with prior work indicating that the stability of spatial frequency preference and orientation preference are independently regulated (Jeon et al., 2018). It would be of interest in future work to examine whether at the level of receptive field sub-structure, the axis of receptive field orientation is preferentially destabilized relative to the size of on-off sub-regions. Alternatively, larger sample sizes could reveal subtle changes in spatial frequency; this would be an indication that instability of spatial frequency contributes to the observed change in signal correlation.

– Please discuss the question of why you were able to elicit relevant visual responses from only a small fraction of V1 neurons imaged for the comparisons made between B1-B2, B2-pDE, etc. What exactly caused this (loss of visual responsiveness, or ability to be fit with the 2D Gaussian model)? How does that impact interpretation of the effects of DE?

As noted in the public view, a consist finding across laboratories is that 30-50% of the neural population in V1 is visually responsive to grating stimuli^1–5^. Regarding the comment on B1-B2 and B2-pDE, we now provide analysis using the pool of neurons that were tracked across all three sessions to alleviant the reviewer’s concern that neurons responsive on only a sub-set of these three sessions drove the main effect (Figure 1 —figure supplement 1E,F). Additional analysis related to Figure 3 is now included that clarifies ~11% of neurons that were initially tuned to grating stimuli on the Baseline 2 imaging session were no longer tuned after dark exposure. Of that 11%, half regained tuning (Figure 3 —figure supplement 1E,F). See also our response to reviewer #2, comment 5.

– The last sentence of the Introduction (lines 104-105) comes out of no-where, stating that "exposure to natural image statistics in the home-cage improves neural discriminability in the adult". It is not clear how the authors arrive at this conclusion, especially here in the Introduction (before methods and results are detailed) and in view of the fact that there is next to no information about the visual environment of their experimental animals in their cages. Were natural image statistics measured? Manipulated? Have the authors established a causal relationship between statistics of natural cage environment and neural tuning or discriminability in their animals or is this speculation? Please clarify. What may additionally help the reader in this context, is for authors to introduce the notion of the natural visual environment earlier, as they describe their experimental manipulations. Light re-introduction is in fact not just re-introduction of light, but re-introduction to a high luminance state + the natural [cage] visual environment.

The sentence was updated to: Furthermore, our results establish that although natural vision, which includes complex scene statistics, has a disrupting influence on tuning stability to simple grating stimuli, natural vision in the home-cage environment improves neural discriminability in the adult.

– Please discuss the temporal requirements for the phenomena observed – or at the very least justify why 8 days were selected for both the DE and the light re-exposure. Would 1 day have sufficed? Would 3? This ties in with my next point ….

8 days of light re-exposure was selected because by design it was necessary to match the duration to the time spend in DE so that we could compute the rate of representational drift (the temporal duration must the same in this case). Text was updated to make this clearer. Given the results, indeed it, would be of interest in future studies to determine how quickly accuracy rebounds upon light re-exposure.

– It would be good to see the discussion of results include some mechanistic hypotheses about the phenomena described. Do the authors think the representational stabilization observed is the result of homeostatic plasticity induced by deprivation or a Hebbian mechanism? What about the effect of visual re-exposure?

The reviewer brings up some interesting questions. We can only speculate. Our working model is that: (1) Under normal natural viewing conditions in the adult, orientation preference is subjected to on-going Hebbian plasticity which has an observable net destabilizing effect on the preference of individual neurons in V1; these changes potentially occupy the null coding space such that they do not impact downstream readout in the visual hierarchy, but are sufficient and necessary to maintain retinotopic and matched organization of receptive field structure. In this scenario, stabilization is not a direct consequence of deprivation-induced homeostasis, rather the removal of a disrupting influence. (2) An underlying ‘backbone’ exists that is resistant to the effects of homeostatic plasticity induced by deprivation, allowing the representation to re-bound back to its original form, despite prominent deprivation-induced homeostatic changes.

– Finally, the authors nicely present their work in the context of the theory proposed by Clopath and colleagues, about the co-existence of neural populations that are more easily perturbed by sensory stimuli and those which are more resistant – the more "stable backbone". While the present work may have confirmed such a scenario in the intact mouse visual cortex, it would be useful to see a discussion of the likelihood that this may also be true in pathological visual systems such as those of amblyopic subjects.

Again, the reviewer brings up some interesting questions. The following is speculative; we welcome the opportunity to include in the manuscript if useful:

The extent to which a stable backbone could be updated to integrate previously deprived input in the case of pathologies such as amblyopia is unclear. Prior work demonstrates that visual perceptual training is effective at improving spatial acuity specifically if the training immediately follows dark exposure (Eaton et al., 2016), such that the rejuvenating effects of dark exposure are still present. It would be of interest in future experiments to determine whether dark exposure transiently allows for a stable backbone to be updated to reflect newly integrated input.

Reviewer #2 (Recommendations for the authors):Despite the significance of the work and the findings, there are a number of points that should be clarified in the manuscript:1. According to supplemental data file 1, 6 mice were recorded in these experiments, 3 aged postnatal day 45-59 and 3 aged postnatal day 83-86. It would be interesting to see if the response to dark exposure was different across the 2 age groups, as predicted by previous work. An experimental timeline would be useful, the above age at imaging initiation is only available in supplemental data.

Thank you for raising this issue. Indeed, the youngest animal (p45) does appear atypical. This information is added in Figure 1 —figure supplement 1A. In addition we rearranged the Table to appear as Table 1, rather than a supplementary file.

2. The representational drift demonstrated in Deitch et al., in visual cortex is not stabilized by the visual stimulation protocol, however Schoonove et al., 2021 show demonstrate that daily exposure to the same odorant stabilizes stimulus selectivity in the olfactory cortex. I'd like to see the authors expand their discussion of these interesting observation.(It would also be more precise for the authors to refer to Schoonove et al., as working in olfactory cortex not association cortex).

Updated to olfactory cortex. We agree that it is possible that representational drift in visual cortex is less sensitive to repeated stimulation presentation compared to olfactory cortex, and it would be of interest to explore. However, the experimental design used in the Deitch et al., study was not designed specifically to test that hypothesis. The stimulus schedule, including which stimuli were presented on a given session and the interval (e.g. days between sessions) at which they were presented differed across mice, and as we understand the Allen Brain data set, it is possible some mice experienced other visual stimuli at different rates. Because of this complication, we prefer to not directly compare Deitch and Schoonove. Note, we did expand the discussion to include some mechanistic hypotheses about the phenomena observed, as suggested by reviewer #1, and on line 272 relate our results to Schoonove et al., 2021.

3. In Figure 1B, “tracked and tuned” neurons were shown in red and “segmented” neurons were shown in yellow. How about “tracked but not tuned” neurons? Were they included in “segmented” group? Since not all visually responsive neurons are tuned, this group should be separated by “segmented”.

The requested Information was added to Figure 1B. We now include blue colorization to indicate neurons tracked but not tuned.

4. In Figure 3F and G, the nature of decrease in tuning should be further discussed. Pacheco et al., 2019 demonstrated, prolonged exposure to darkness increases the firing rates upon LRx. Is the decrease in reliability due to the decrease in SNR? Any difference in firing rate changes between “neurons remained tune" vs "neurons lost tunin”?

Thank you for this thoughtful comment, indeed given prominent changes in immediate early gene expression changes in basal firing rate could be significant. We added additional analysis of basal activity to address this issue and did not detect an increase in basal activity in the post-DE imaging session (Figure 3 —figure supplement 1D). Note, the observed increase in firing by Pacheco et al., 2019 was transient, peaking at 10 minutes and returning to baseline within 30-60 minutes. In our case, recording was initiated 30-40 minutes after the transition from dark to low-light during animal positioning and neuron tracking, thus is consistent with Pacheco et al., 2019.

5. Unless those 30% of neurons that lost tuning after pDE are the ones regain tuning, visual representation after 8 d LRx is different from pre-DE condition (B1 and B2). It is not clear if the identity of 30% of neurons were tracked after 8 d LRx to prove this.

Apologies for the confusion. In Figure 3, all analysis was performed on the same tracked pool of neurons; neurons were segmented in all 4 sessions: Baseline 1 (B1), Baseline 2 (B2), Post-DE (pDE), and Recovery (Rec). We are not entirely sure what the reviewer means by ‘those 30% of neurons that lost tuning after pDE’. To make this section clearer, we quantified the fraction of neurons that were tuned and found that on average across the 6 mice there was an 11±4% decrease in tuned neurons from Baseline 2 to Post-DE. Of the 11% that were lost, 6 % were re-gained (new Figure supplement 2E,F).

Reviewer #3 (Recommendations for the authors):The paper uses a combination of analysis methods that are both thoughtful and complementary. My main recommendations are for additional analyses to strengthen the findings.(1) Please mention whether the monocular zone of V1 or the binocular zone of V1 was imaged. This is relevant for comparison with monocular deprivation studies.

This information is now provided in Figure 1 —figure supplement 2.

(2) The finding that a fraction of neurons show more stable orientation preferences following dark exposure depends on the quality of the fits for the tuning curves. While the authors present a control where they remove 10% of neurons with the worst R2 from their analysis, and show that the conclusions still hold, other complementary analysis can be used. Does R2 depend on change in orientation preference? If neurons were to be ranked according to their change in orientation preference and split into three equal groups, is there a difference in R2 between the three groups?

We agree that in order for our conclusion to be valid, there cannot be a systematic difference in the quality of fits across imaging sessions, and that complementary analysis is useful. We include three different (complementary) analyses that address this issue. First, our analysis of three groups, B1, B2, and pDE, demonstrates that there is no difference in the goodness of fit values across imaging sessions (Figure 1 —figure supplement 1B); this is direct evidence that there is not a systematic difference in quality of fits underlies the observed difference reported in Figure 1D. Second, removing the worst 10% did not impact the results (Figure 1 —figure supplement 1C). Third, stability of signal correlation was analyzed for all responsive neurons, regardless of tuning (Figure 1 —figure supplement 1D). The reviewer suggests a forth analysis, however a priori, one might expect that there is a correlation between the stability of orientation preference and R2 value. This could have a biological basis. For example, higher trial-to-trial variability could be predictive that a neuron will be more likely to shift its orientation preference. Such a correlation does not invalidate the conclusion, but does mean that the first analysis described above is essential. Note, the criteria for including specific neurons was independent of the R square value, we used a permutation test to determine whether a neuron was tuned or not.

In case helpful, we include a version of the requested analysis in this rebuttal (Author response image 1). Note, the exact requested analysis was not possible because there are two R2 values associated with each change in orientation value.

**Author response image 1. sa2fig1:** Analysis of R squared values. The change in orientation preference for all neurons included in Figure 1D were sorted and divided into three equal groups (n = 159 neurons, ‘Top’, ‘Middle’, and ‘Lower’) and the two corresponding R square values generated from the 2-dimentional Gaussian fits for each neuron were averaged to give one value per neuron. The mean and standard deviation of the R square values are plotted; the values are largely overlapping. P values are calculated using the Wilcoxon rank-sum test and are not corrected for multiple comparisons. As expected, there is a correlation between orientation stability and the R square value. Importantly, the analysis presented in Figure 1 —figure supplement 1B indicates that the distribution of R square values is similar across conditions.

(3) The paper pools visual responses across six mice to show a smaller change in orientation preference following dark exposure compared to baseline. Is there an effect if you do Wilcoxon rank sum test on visual responses from each individual mouse (ie. Figure 1D statistical test performed on visual responses from individual mice)?

This comment is related to reviewer #2 comment #1. We now provide statistical analysis on individual animals in Figure 1 —figure supplement 1A. From this analysis we can conclude that the effect is not preferentially driven by one or two animals. We did find one atypical animal; a possible explanation is a difference in age, which is now noted in the text. To make sure the data are clear to the reviewer, in addition to the new summary plots, we provide the full cumulative plots of each animal in the rebuttal (Author response image 2).

**Author response image 2. sa2fig2:** The change orientation preference of individual animals that comprises Figure 1D. Blue line Δ Control, Red line Δ DE. The number of neurons on an individual animal basis is too small to make statistical inferences. Note, the trend is present in 5 of 6 mice. See also Figure 1 —figure supplement 1A.

(4) Additional analysis is needed to support the following sentence in the Abstract or this sentence should be softened: "a decrease in response reliability across a broad range of spatial frequencies accounted for the disruption". The paper shows that there is both decreased reliability and degraded classifier performance. However, to address whether decreased reliability indeed degrades classifier performance, the authors could remove the least reliable neurons and compare classifier performance (the comparison may have to be with a matched number of neurons).

Agreed. The sentence was updated to: "a decrease in response reliability across a broad range of spatial frequencies likely contributed to the disruption".